# A Semi-Supervised Machine Learning Approach in Predicting High-Risk Pregnancies in the Philippines

**DOI:** 10.3390/diagnostics12112782

**Published:** 2022-11-14

**Authors:** Julio Jerison E. Macrohon, Charlyn Nayve Villavicencio, X. Alphonse Inbaraj, Jyh-Horng Jeng

**Affiliations:** 1Department of Information Engineering, I-Shou University, Kaohsiung City 84001, Taiwan; 2College of Information and Communications Technology, Bulacan State University, Bulacan 3000, Philippines

**Keywords:** maternal health, high-risk pregnancy, semi-supervised learning, machine learning, prediction model

## Abstract

Early risk tagging is crucial in maternal health, especially because it threatens both the mother and the long-term development of the baby. By tagging high-risk pregnancies, mothers would be given extra care before, during, and after pregnancies, thus reducing the risk of complications. In the Philippines, where the fertility rate is high, especially among the youth, awareness of risks can significantly contribute to the overall outcome of the pregnancy and, to an extent, the Maternal mortality rate. Although supervised machine learning models have ubiquity as predictors, there is a gap when data are weak or scarce. Using limited collected data from the municipality of Daraga in Albay, the study first compared multiple supervised machine learning algorithms to analyze and accurately predict high-risk pregnancies. Through hyperparameter tuning, supervised learning algorithms such as Decision Tree, Random Forest, Support Vector Machine, K-Nearest Neighbors, Naïve Bayes, and Multilayer Perceptron were evaluated by using 10-fold cross validation to obtain the best parameters with the best scores. The results show that Decision Tree bested other algorithms and attained a test score of 93.70%. To address the gap, a semi-supervised approach using a Self-Training model was applied to the modified Decision Tree, which was then used as the base estimator with a 30% unlabeled dataset and achieved a 97.01% accuracy rate which outweighs similar studies.

## 1. Introduction

Maternal health is one of the key priorities of the United Nations. Ever since the Millennium Development Goals were created back in 2000 up to its deadline in 2015, and from the transition to the Sustainable Development Goals in 2015 up to today, the overarching goal is still there, although there has been a significant reduction but not full eradication [1].

In this modern world, most, if not all, pregnancy complications can be treated or even prevented. However, according to the World Health Organization, approximately 810 women die every day from preventable causes related to pregnancy and childbirth and 94% of these deaths occur in low and middle-income countries with teens and young adolescents less than twenty years old face a higher risk than other women [2]. The problem lies with the awareness and access of a safe and clean medical facility as well as frequent or regular check-ups during the course of the pregnancy. Only then using these methods, especially during regular check-ups, expectant mothers will then be aware of their situation such as being tagged as a high-risk pregnancy.

High-risk pregnancy involves an increased health risk for the mother, unborn baby, or both. Although all pregnancies carry risks, a high-risk pregnancy may need extra care before, during, and after the pregnancy [3]. Moreover, some pregnancies become high-risk as they progress, while some women are at an increased risk for complications even before they become pregnant for a variety of reasons. Thus, early and regular prenatal care helps many women have healthy pregnancies and deliveries without complications [4].

The purpose of this study was to (1) make use of the data collected from local health centers and predict the risk of pregnancy complications; (2) make use of the prediction to fully prepare not just the mother but also the community in the successful delivery of the child [5]; and (3) leverage the use of semi-supervised approach in boosting the accuracy of supervised machine learning models.

In this paper, the authors compare the machine learning algorithms and methodology to those of other papers, as seen in Section 2. Then they explain the step-by-step approach in the processing of data gathered in Section 3, of which the results are then displayed and discussed in Section 4, together with a discussion in Section 5, and the authors’ conclusions and recommendations appear in Section 6.

This study used a semi-supervised approach in machine learning, which involved the use of the supervised machine learning models and augmenting the said model, especially in some circumstances when data are weak or where vast quantities of data are unlabeled [6,7].

## 2. Related Studies

Multiple machine learning algorithms and methods were used to predict different types of diseases, from diabetes to COVID-19 [8]. The ubiquity of mobile phones [9], specifically their use in everyday life, presents a unique opportunity. Using such existing systems makes it easier to access digital tools in maternal health programs for the real-time implementation of predictive models [10]. Data capturing has been one of the most challenging steps [11] in deploying these new tools. However, over time, systems have been in place to provide a simpler way for the eventual use to build prediction models for things such as for pre-eclampsia [12], delivery location [10], risk factors associated with Caesarean section [13], and neonatal infections [14].

As for high-risk pregnancies, some studies used supervised machine learning algorithms such as the J48 Decision Tree (J48 DT), Random Forest (RF), Support Vector Machine (SVM), K-Nearest Neighbors (KNN), and Naïve Bayes (NB), to name a few.

For the study by Lakshmi et al., they utilized the J48 DT or C4.5 Decision Tree to predict the risk of complications arising in the period of gestation, using twelve parameters, such age, parity, history of pre-eclampsia and gestational diabetes, blood pressure, and weight gain, among others, and classified the data as being high, mid, or no risk. Out of 230 data samples, 152 were correctly classified or about 71.30% accuracy rate.

In the study of Akbulut et al., they used nine different algorithms, namely Averaged Perceptron (AP), Boosted Decision Tree (BDT), Bayes Point Machine (BPM), Decision Forest (DF), Decision Jungle (DJ), Locally Deep Support Vector Machine (L-DSVM), Logistic Regression (LR), Neural Network (NN), and Support Vector Machine (SVM). Using the data gathered from their created Android Mobile Application, the Decision Forest model obtained the highest accuracy of 89.50%.

Using the same dataset as Akbulut, Bautista et al. compared only four algorithms, Decision Tree (DT), Random Forest Decision Tree (RFDT), K-Nearest Neighbors (KNN), and Support Vector Machine (SVM). For the training of the said algorithms, the Random Forest Decision Tree (RFDT) algorithm obtained the highest accuracy score of 90.00. They then used the said model to create an Android Mobile Application.

Lastly, Ahmed et al. used the IoT for the generation of data with eight algorithms, namely a Decision Tree (DT), Random Forest (RF), Support Vector Machine (SVM), Sequential Minimal Optimization (SMO), Logistic Regression (LR), Naïve Bayes (NB), IBk, and Logistic Model Tree (LMT), as well as a 15-fold cross-validation, in training the models. Ultimately, the Decision Tree (DT) algorithm, using *GridSearchCV* (for hyperparameter tuning), gave the highest accuracy rate, i.e., 97.00%. Below is a comparison of all the similar studies conducted, including the proposed method of the authors.

Table 1 above compares not just similar studies but also their machine learning methodology, such as algorithms and accuracy obtained. It can be observed that, although most, if not all, of the best algorithms are a Decision Tree or an alteration of it, not all data and methodologies are similar. Knowing that the supervised machine learning models have reached their highest potential accuracy, the gap exists to find a way to further improve its accuracy. Therefore, using a semi-supervised approach can be highly beneficial in providing additional training examples to boost the prediction accuracy of supervised models [15].

## 3. Methodology

The machine learning process was used as the methodology for this project, but some parts deviated in order to include other added features, such as hyperparameter tuning and data balancing. The whole process is described in the block diagram in Figure 1 below.

The process involved five phases: data collection, data processing, hyperparameter tuning, comparative analysis, and modeling. This then generates the high-risk pregnancy prediction model to be used. The method involved the use of Python as the main programming language, with dependencies such as *Numpy*, *Pandas*, and *Sci-Kit Learn*, on which the algorithms were based. Using the author’s Windows-based personal computer with an Intel i7 10th generation processor and NVIDIA GeForce GTX 1650 Ti graphics card, the models were run by using IDLE Extensions for Python (IDLEx) as the main IDE for this study.

### 3.1. Data Collection

Firstly, the data were collected from the *Barangay* (Village) of *Anislag*, located in the municipality of *Daraga*, *Albay*, situated in the eastern part of *Luzon Island* in the Philippines. The dataset came from the village’s Pre-Natal Target Client List (TCL) from the local health center, which has twelve columns in total, and the authors considered only seven of those as predictors and one as the label attribute. The names and descriptions of the attributes can be viewed in Table 2. The initial label attributes were annotated by a registered midwife from the Philippines. The said dataset, which was used as training data, contained 90 rows belonging to two risk classifications: high risk, with 56 rows; and low risk, with 34 rows. The said data were then split into a 70–30 training–testing dataset.

In the municipality of Daraga in the province of Albay, the Pre-Natal TCL of the village includes the following demographics in Table 3 that they catered to during the data-collection phase.

With a total of 90 respondents, the average age (in years) that they have catered to is 26.87 years old, with the youngest being 15 years old and the oldest being 45 years old. The Gravida, or the number of times the woman has been pregnant, for the village averaged to about 3 pregnancies, with a minimum of 1 and maximum of 5 pregnancies. The parity, or the number of times the woman has given birth (whether dead or alive), averages to 2, with some women having 0 or their first pregnancy (they never have given birth yet) and a maximum of 4 births. The height of women in Daraga averages to about 154.99 cm, with the shortest being 140.5 cm and tallest being 190 cm. Women’s weight in Daraga averages to about 56.9 kg, with the lightest being 39 kg and heaviest being 88 kg. The number of visits to the village’s health center averaged to about 3 visits, with the minimum being 1 and maximum being 5 visits.

Due to the limited data, the authors used the Synthetic Data Vault (SDV) to generate synthetic data. The SDV generates synthetic data by applying mathematical techniques and machine learning models, such as the deep learning model. Several studies have embraced the use of synthetic data, which can be applied for real-world applications [21]. An additional 900 new rows were generated for testing.

### 3.2. Data Processing

Due to the modest number of data gathered, data balancing was performed, as it is critical for promoting a balanced prediction rate. Through the use of the Synthetic Minority Oversampling Technique (SMOTE), the original 90 rows are now 112, with both 56 high- and low-risk classification as seen in Figure 2.

This is the same for the synthetic data [21], as seen in Figure 3, that were generated; the created 900 rows were also unbalanced. After using SMOTE, 900 rows are now 1270, with 635 equally classified values.

### 3.3. Hyperparameter Tuning

In order to obtain the best parameters for each of the supervised machine learning algorithm, hyperparameter tuning was performed by using *GridSearchCV*. This model, together with a 10-fold cross-validation test, generated the best parameter to be used together with the best test scores, using training data. This is crucial in ensuring the most accurate prediction result. Following this, below are the six supervised machine learning algorithms used.

#### 3.3.1. Decision Tree (DT)

The goal in using the Decision Tree classifier is to create a training model that predicts the class or value of the target variable by learning simple decision rules inferred from training data. In Decision Trees, for predicting a class label, we start from the root of the tree and compare the values of the root attribute with the record’s attribute. On the basis of comparison, we follow the branch corresponding to that value and jump to the next node, and so on [22].

#### 3.3.2. Random Forest (RF)

Random Forest, as its name implies, consists of a large number of individual decision trees that operate as an ensemble. Each individual tree in the Random Forest spits out a class prediction, and the class with the most votes becomes the model’s prediction. In data science, the reason why the Random Forest works well is that “a large number of relatively uncorrelated models (trees) operating as a committee will outperform any of the individual constituent models” [23].

#### 3.3.3. Support Vector Machine (SVM)

The idea of the Support Vector Machine is simple: The algorithm creates a line or a hyperplane, which separates the data into classes. The SVM finds the points closest to the line from both the classes; these points are called support vectors. It then computes the distance between the line and the support vectors, of which the distance is called the margin. The goal is to maximize the margin. The hyperplane for which the margin is maximum is the optimal hyperplane. Thus, the SVM tries to decide the boundary in such a way that the separation between the two classes is as wide as possible [24].

#### 3.3.4. K-Nearest Neighbors (KNN)

For the K-Nearest Neighbors classification [25], the goal is to predict the target class label as the class label that is most often represented among the *k* most similar training examples for a given query point. In other words, the class label can be considered as the “mode” of the k training labels or the outcome of a “plurality voting”. Note that, in the literature, KNN classification is often described as a “majority voting”. While the authors usually mean the right thing, the term “majority voting” is a bit unfortunate, as it typically refers to a reference value of >50% for making a decision. In the case of binary predictions (classification problems with two classes), there is always a majority or a tie. Hence, a majority vote is also automatically a plurality vote. However, in multi-class settings, we do not require a majority to make a prediction via KNN. For example, in a three-class setting, a frequency > 1/3 (approximately 33.3%) could already be enough to assign a class label [26].

#### 3.3.5. Naïve Bayes (NB)

Naïve Bayes is a probabilistic machine learning algorithm based on the Bayes Theorem, used in a wide variety of classification tasks. Bayes’ Theorem, on the other hand, is a simple mathematical formula used for calculating conditional probabilities, and conditional probability is a measure of the probability of an event occurring given that another event has (by assumption, presumption, assertion, or evidence) occurred. The following formula is used:
(1)$$P(A|B)=\frac{P(B|A)\cdot P(A)}{P(B)}$$

This tells us that P(A|B) is how often A happens given that B happens, also called posterior probability; P(B|A) is how often B happens given that A happens; P(A) is how likely A is on its own; and how likely B is on its own is written as P(B). In simpler terms, Bayes’ Theorem is a way of finding a probability when we know certain other probabilities [27].

#### 3.3.6. Multilayer Perceptron (MLP)

Multilayer Perceptron is a Neural Network that learns the relationship between linear and nonlinear data. As seen in Figure 4, MLP has input and output layers, and one or more hidden layers with many neurons stacked together. Moreover, while in the Perceptron, the neuron must have an activation function that imposes a threshold, such as ReLU or sigmoid, neurons in an MLP can use any arbitrary activation function [28].

### 3.4. Comparative Analysis

A comparative analysis was performed to evaluate the performances of the different supervised machine learning algorithms by using a 10-fold cross-validation testing, using testing data, and using the following criteria as a basis for comparison [8].

#### 3.4.1. Accuracy

Accuracy is the measurement of all the correctly predicted instances over the total predictions made by the model; it computes the ratio of the correctly classified samples which are true positives (TPs) and true negatives (TNs) over the total number of predictions, which includes the TPs, TNs, and misclassified predictions, such as false positives (FPs) and false negatives (FNs). The formula for accuracy can be seen in the equation below:


(2)
$$\mathrm{Accuracy}=\frac{\mathrm{TP}+\mathrm{TN}}{\mathrm{TP}+\mathrm{TN}+\mathrm{FP}+\mathrm{FN}}$$


#### 3.4.2. Sensitivity

Sensitivity is the ratio of correctly classified high-risk samples to all numbers of high-risk samples in the dataset. The sensitivity of the classifier is also known as the true-positive Rate (TPR), and it can be computed by using the following formula:
(3)$$\mathrm{Sensitivity}=\frac{\mathrm{TP}}{\mathrm{TP}+\mathrm{FN}}$$

#### 3.4.3. Specificity

Specificity is the proportion of the low-risk samples that were correctly classified as low-risk, which is also termed as the true-negative rate (TNR). The specificity score can be computed by using the following formula:
(4)$$\mathrm{Specificity}=\frac{\mathrm{TN}}{\mathrm{TN}+\mathrm{FP}}$$

### 3.5. Modeling

After comparing all six algorithms, they were used as a base estimator for the Self-Training model. By using semi-supervised learning, the authors leveraged the limited amount of data without labels to obtain the best accurate result. Semi-supervised learning is a situation in which the training data of some of the samples are not labeled. These algorithms can perform well when they have a very small number of labeled points and a large number of unlabeled points [29].

Using the Self-Training model, a given supervised classifier (base estimator) can function as a semi-supervised classifier, allowing it to learn from unlabeled data. In each iteration, the base estimator predicts labels for the unlabeled samples and adds a subset of these labels to the labeled dataset. For the training of the Self-Training model, the newly created balanced synthetic data were used and were trained to decipher a 30% unlabeled dataset in order to test out its accuracy.

## 4. Results

Using the abovementioned methodology, the study was performed, and the results are presented in this section. Using training data, the best parameters from the hyperparameter tuning were then used and are presented in Section 4.1. We then used those best parameters to train the model and used testing data compared with the other supervised machine learning models to obtain the best accuracy. The algorithm with the best accuracy was then used for the Self-Training model, which then used the balanced synthetic data with a 30% unlabeled dataset.

### 4.1. Results of Hyperparameter Tuning

Using the *GridSearchCV* method with a 10-fold cross-validation, here are the results of the hyperparameter tuning. The highlighted rows are the best parameters for each algorithm, as compared to their mean score (score), after completing a 10-fold cross-validation.

#### 4.1.1. Decision Tree (DT)

For the DT algorithm, the researchers used the criterion, minimum samples in node, and the maximum depth for the parameters. The results are displayed in Table 4 below.

In Table 4, the top ten combinations of criterion, minimum samples in node, and the maximum depths as parameters are shown with rank-one combination as the entropy criterion, five minimum samples in the node, and a ten for maximum depth, giving a mean score of 90.08% accuracy.

#### 4.1.2. Random Forest (RF)

For the RF algorithm, the researchers used the criterion, minimum samples, maximum depth, number of estimators, and bootstrap for the parameters. The results are displayed in Table 5 below.

In Table 5, the top ten combinations of criterion, minimum samples, maximum depth, number of estimators, and bootstrap as parameters are shown with rank-one combination as entropy criterion, 5 minimum samples, a 10 maximum depth, 200 estimators, and bootstrap being true, giving a mean score of 93.71% accuracy.

#### 4.1.3. Support Vector Machine (SVM)

For the SVM algorithm, the researchers used the C, degree, kernel, and gamma for the parameters. The results are displayed in Table 6 below.

In Table 6, the top ten combinations of C, degree, kernel, and gamma as parameters are shown with the rank-one combination as a C of 5, degree of 3, a Radial Basis Function kernel, and gamma of 0.1, giving a mean score of 97.27% accuracy.

#### 4.1.4. K-Nearest Neighbors (KNN)

For the KNN algorithm, the researchers used the metric, number of neighbors, and the weights for the parameters. The results are displayed in Table 7 below.

In Table 7, the top ten combinations of metric, number of neighbors, and the weights as parameters are shown with rank-one combination as the cosine metric, seven neighbors, and distance as weight, giving a mean score of 95.53% accuracy.

#### 4.1.5. Naïve Bayes (NB)

For the NB algorithm, the researchers used only var smoothing for the parameters. The results are displayed in Table 8 below.

In Table 8, the top ten var smoothing as parameters are shown with rank-one var smoothing of 0.151991, giving a mean score of 83.79% accuracy.

#### 4.1.6. Multilayer Perceptron (MLP)

For the MLP algorithm, the researchers used hidden layer sizes, activation, solver, alpha and the learning rate for the parameters. The results are displayed in Table 9 below.

In Table 9, the top ten combinations of hidden layer sizes, activation, solver, alpha, and the learning rate as parameters are shown with rank-one combination as (50, 100, 50) hidden layer size, tanh activation, adam solver, alpha of 0.0001, and a constant learning rate giving a mean score of 98.18% accuracy.

### 4.2. Comparative Analysis Results

Using the top or best parameters for each of the supervised machine learning algorithm, we fit the model and used the testing data. Through this, the following accuracies, sensitivities, and specificities were produced.

From Table 10, we see that, with the hyperparameter tuning, the lowest in terms of accuracy is the Naïve Bayes algorithm, having only a 72.18% accuracy, with a 67.02% sensitivity and 77.37% specificity. Next is the K-Nearest Neighbors, with 84.25% accuracy, 87.43% sensitivity, and 81.05% specificity. Random Forest had a 90.29% accuracy, 94.24% sensitivity, and 86.32% specificity. On the other hand, the Support Vector Machine had a 90.81% accuracy, 94.24% sensitivity, and 87.37% specificity. The Multilayer Perceptron had one of the highest accuracies, at 91.86%, with 91.10% sensitivity and a 92.63% specificity. Lastly, the Decision Tree algorithm had the best score in terms of accuracy, sensitivity, and specificity, at 93.70%, 90.05%, and 97.37%, respectively. This correlates to the related systems where the Decision Tree algorithm was used and was the best among others.

### 4.3. Self-Training Model Performance Result

As the authors used a different approach to predict high-risk pregnancies, modeling the Self-Training algorithm together with the modified Decision Tree algorithm was used and yielded the following results.

By using the Decision Tree algorithm as the base estimator for the Self-Training model, Table 11 indicates a 97.01% accuracy rate, using a 30% unlabeled dataset. With 609 rows as true positive (TP), 12 rows as false positive (FP), 26 rows as false negative (FN), and 623 rows as the true negative (TN). The precision rates for the high- and low-risk pregnancies are 98.07% and 95.99%, respectively, while the recall rates for the high- and low-risk pregnancies are 95.91% and 98.11%, respectively. Since the Self-Training model works better with an unlabeled dataset, 30% of the test data were randomly unlabeled to obtain a 635–635 or balanced split between high- and low-risk pregnancies.

## 5. Discussion

This paper was designed to further increase, if not augment, the already ubiquitous and widely used supervised machine learning algorithm. Multiple studies have already been performed to further increase the classification accuracy. However, limitations on data, such as having too much unlabeled data or weak data, can significantly affect the model. With this approach, our goal was to answer the data gap and to apply it to something important and worthwhile, such as high-risk pregnancies, which can affect even maternal mortality. As the Philippines is a middle-income country without many resources, using a simpler, easier, and affordable option such as this model can be significant, as it can be deployed to low-income or even geographically isolated and disadvantaged areas (GIDAs) [5].

## 6. Conclusions

This study aimed to build a high-risk pregnancy predictor model by evaluating six supervised machine learning algorithms, namely Decision Tree, Random Forest, Support Vector Machine, K-Nearest Neighbors, Naïve Bayes, and Multilayer Perceptron. A comparative analysis was made by evaluating the models’ performance in a 10-fold cross-validation, using the Python programming language. The results show that a modified Decision Tree algorithm with entropy criterion, 5 minimum samples in the node, and a 10 maximum depth was the best machine learning algorithm, with an accuracy rate of 93.70%, sensitivity rate of 90.05%, and a specificity rate of 97.37%. The authors then used this algorithm as the base estimator for the Self-Training model and used 30% unlabeled data for testing, thereby obtaining a 97.01% accuracy rate. The best model corresponds with related studies having a modified Decision Tree as the best algorithm for high-risk pregnancies. Moreover, our findings also show that the Multilayer Perceptron is the second-best algorithm being considered in the creation of the predictor. Following this are the Support Vector Machine and Random Forest as the third and fourth best algorithms considered. Lastly, K-Nearest Neighbors and Naïve Bayes were the bottom two algorithms, which fared low in terms of accuracy.

With this, the study can be a basis for a decision support system for our midwives and health workers. Due to the poor conditions and lack of available resources of these health workers, using this model can greatly reduce their workload in terms of assessing their patient’s risk level. Moreover, mothers can personally use this model to know their risk level, should they be able to access this platform. In future studies, the authors plan to develop an easy-to-use web application [30], not just for health workers, but also for mothers to eventually use but with medical guidance. This application will then tell the mothers their next steps and will alert health workers to their risk level and condition.

## Figures and Tables

**Figure 1 diagnostics-12-02782-f001:**
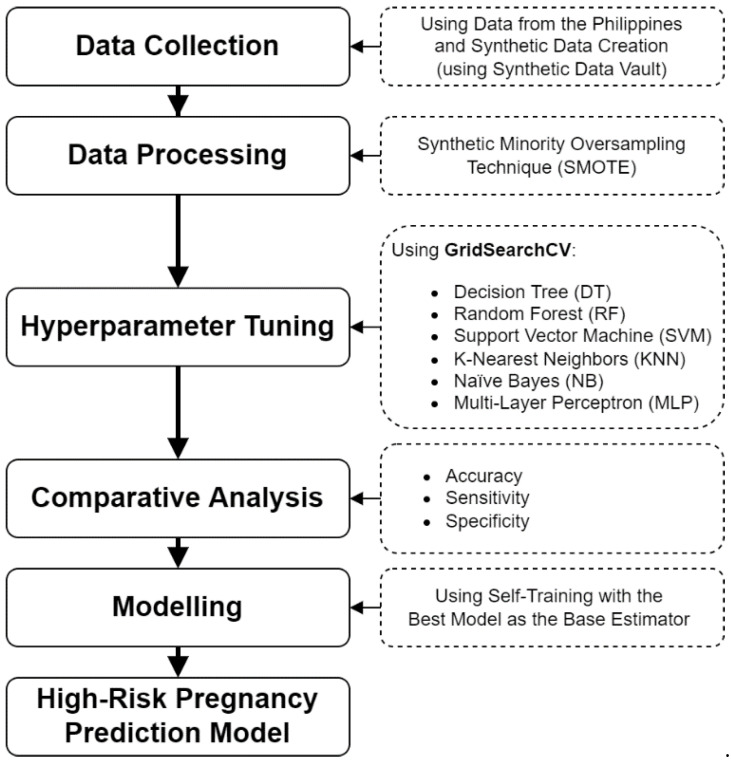
Block diagram of the machine learning process.

**Figure 2 diagnostics-12-02782-f002:**
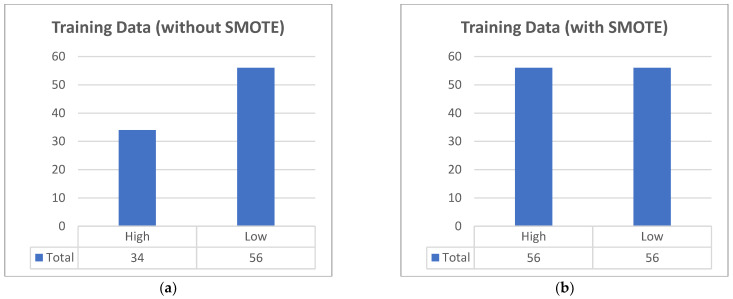
The dataset collected was unbalanced. (**a**) Originally, there were 90 rows; there were more low-risk pregnancies than high-risk. (**b**) Using SMOTE, both classifications are now equally 56 rows, totaling 112 rows.

**Figure 3 diagnostics-12-02782-f003:**
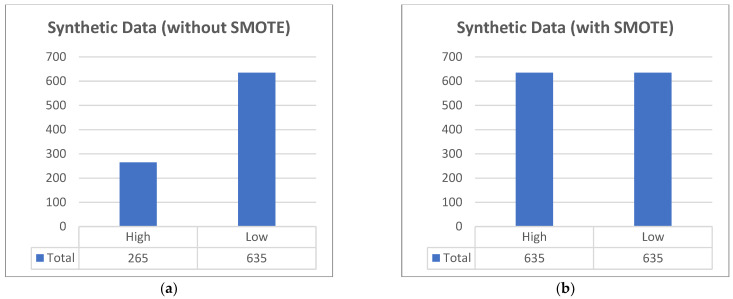
The synthetic data generated were also unbalanced. (**a**) Originally, there were 900 rows; there were more low-risk pregnancies than high-risk. (**b**) Using SMOTE, both classifications are now equally 635 rows, totaling 1270 rows.

**Figure 4 diagnostics-12-02782-f004:**
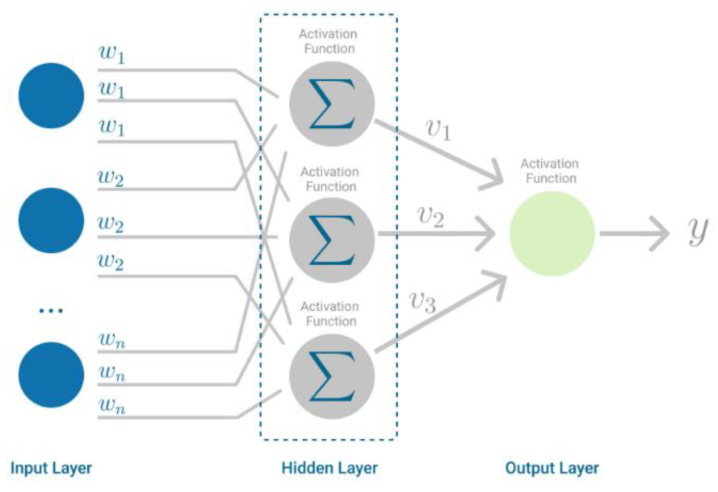
Multilayer Perceptron.

**Table 1 diagnostics-12-02782-t001:** The comparison of related studies conducted for the prediction of high-risk pregnancies.

Study	Machine Learning Algorithms	Best Algorithm	Accuracy Obtained	Reference
Lakshmi, et al.	J48 DT	J48 DT	71.30%	[16]
Akbulut, et al.	AP, BDT, BPM, DF, DJ, L-DSVM, LR, NN, SVM	DF	89.50%	[17]
Bautista, et al.	DT, RFDT, KNN, SVM	RFDT	90.00%	[18]
Ahmed, et al.	DT, RF, SVM, SMO, LR, NB, IBk, LMT	DT	97.00%	[19]
Proposed Method	DT, RF, KNN, SVM, NB, MLP	DT with Self-Training	97.01%	

**Table 2 diagnostics-12-02782-t002:** Dataset attribute (predictors and class label), data types, and description.

Attribute	Type	Description
Age	Nominal	Age (in years) when the woman is pregnant.
Gravida	Nominal	The number of times a woman has been pregnant [20].
Parity	Nominal	The number of times a woman has given birth with a gestational age of 24 weeks or more, regardless of whether the child was born alive or was stillborn [20].
Height	Nominal	Head-to-toe measurement in centimeters (cm).
Weight	Nominal	Body mass measurement in kilograms (kg).
Trimester	Nominal	The stage (out of three trimesters) during the pregnancy in which the woman visited the health center.
Visits	Nominal	Number of times the woman visited the health center.
Risk Level	Class Label	Classified as either high-risk or low-risk pregnancy.

**Table 3 diagnostics-12-02782-t003:** Demographic data of the municipality of Daraga, Albay.

Attribute	Type	Value
Age (years)	Total	90
	Mean	26.87
	Minimum	15
	Maximum	45
Gravida	Mean	3
	Minimum	1
	Maximum	5
Parity	Mean	2
	Minimum	0
	Maximum	4
Height (cm)	Mean	154.99
	Minimum	140.5
	Maximum	190
Weight (kg)	Mean	56.9
	Minimum	39
	Maximum	88
Visits	Mean	3
	Minimum	1
	Maximum	5

**Table 4 diagnostics-12-02782-t004:** Decision Tree hyperparameter-tuning results.

No.	Criterion	Min Samples in the Node	Maximum Depth	Score	Ranking
**1**	**entropy**	**5**	**10**	**90.08**	**1**
2	entropy	5	5	90.08	1
3	entropy	5	20	90.08	1
4	entropy	10	5	89.17	4
5	entropy	10	3	89.17	4
6	entropy	10	2	89.17	4
7	entropy	5	2	89.17	4
8	entropy	10	10	89.17	4
9	gini	5	10	89.17	4
10	gini	5	5	89.17	4

**Table 5 diagnostics-12-02782-t005:** Random Forest hyperparameter-tuning results.

No.	Criterion	Min Samples	Max Depth	No. of Estimators	Bootstrap	Score	Ranking
**1**	**entropy**	**5**	**10**	**200**	**True**	**93.71**	**1**
2	entropy	5	5	300	False	93.71	1
3	entropy	5	5	100	False	93.71	1
4	gini	5	5	200	True	93.71	1
5	entropy	5	20	300	True	93.71	1
6	gini	5	10	100	True	92.80	6
7	gini	5	10	200	True	92.80	6
8	gini	5	10	100	False	92.80	6
9	entropy	5	10	100	True	92.80	6
10	entropy	5	5	200	False	92.80	6

**Table 6 diagnostics-12-02782-t006:** Support Vector Machine hyperparameter-tuning results.

No.	C	Degree	Kernel	Gamma	Score	Ranking
**1**	**5**	**3**	**Radial Basis Function**	**0.1**	**97.27**	**1**
2	5	1	Radial Basis Function	0.1	97.27	1
3	2	2	Radial Basis Function	0.1	97.27	1
4	2	1	Radial Basis Function	0.1	97.27	1
5	2	3	Radial Basis Function	0.1	97.27	1
6	5	2	Radial Basis Function	0.1	97.27	1
7	10	2	Radial Basis Function	0.1	96.36	7
8	10	1	Radial Basis Function	0.1	96.36	7
9	10	3	Radial Basis Function	0.1	96.36	7
10	1	2	Radial Basis Function	0.1	95.45	10

**Table 7 diagnostics-12-02782-t007:** K-Nearest Neighbors hyperparameter-tuning results.

No.	Metric	Neighbors	Weights	Score	Ranking
**1**	**cosine**	**7**	**distance**	**95.53**	**1**
2	manhattan	5	distance	95.45	2
3	l1	9	distance	95.45	2
4	cityblock	5	distance	95.45	2
5	cityblock	9	distance	95.45	2
6	l1	5	distance	95.45	2
7	manhattan	9	distance	95.45	2
8	l2	3	distance	94.62	8
9	euclidean	3	distance	94.62	8
10	nan_euclidean	3	distance	94.62	8

**Table 8 diagnostics-12-02782-t008:** Naïve Bayes hyperparameter-tuning results.

No.	Var Smoothing	Score	Ranking
**1**	**0.151991**	**83.79**	**1**
2	0.123285	83.79	1
3	0.1	83.79	1
4	0.065793	82.88	4
5	0.081113	82.88	4
6	0.015199	80.30	6
7	0.018738	80.30	6
8	0.008111	80.30	6
9	0.035112	80.15	9
10	0.187382	80.15	9

**Table 9 diagnostics-12-02782-t009:** Multilayer Perceptron hyperparameter-tuning results.

No.	Hidden Layer Sizes	Activation	Solver	Alpha	Learning Rate	Score	Ranking
**1**	**(50, 100, 50)**	**tanh**	**adam**	**0.0001**	**constant**	**98.18**	**1**
2	(50, 100, 50)	tanh	adam	0.0001	adaptive	98.18	1
3	(50, 50, 50)	relu	adam	0.0001	constant	97.35	3
4	(50, 100, 50)	tanh	adam	0.05	constant	97.95	3
5	(50, 50, 50)	tanh	adam	0.0001	constant	97.27	5
6	(50, 50, 50)	tanh	adam	0.05	constant	96.44	6
7	(100,)	tanh	adam	0.05	adaptive	95.53	7
8	(100,)	tanh	adam	0.05	constant	95.53	7
9	(50, 50, 50)	relu	adam	0.05	adaptive	95.53	7
10	(100,)	tanh	adam	0.0001	adaptive	95.53	7

**Table 10 diagnostics-12-02782-t010:** Comparative analysis.

Algorithm	Accuracy	Sensitivity	Specificity
**Decision Tree**	**93.70%**	**90.05%**	**97.37%**
Random Forest	90.29%	94.24%	86.32%
Support Vector Machine	90.81%	94.24%	87.37%
K-Nearest Neighbors	84.25%	87.43%	81.05%
Naïve Bayes	72.18%	67.02%	77.37%
Multilayer Perceptron	91.86%	91.10%	92.63%

**Table 11 diagnostics-12-02782-t011:** Confusion matrix.

Decision Tree with Self-Training	Predicted High Risk	Predicted Low Risk	Recall
Actual High Risk	609	26	95.91%
Actual Low Risk	12	623	98.11%
Precision	98.07%	95.99%	

## Data Availability

Not applicable.

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
