# Peer review of "A Semi-Supervised Machine Learning Approach in Predicting High-Risk Pregnancies in the Philippines"

_diagnostics, 2022, doi:10.3390/diagnostics12112782_

Round 1
Reviewer 1 Report
The title of the paper does not correspond to the content of the paper. The work use only supervised machine learning algorithms.
The paper make use of data collected from local Health Centers and predict the risk of pregnancy complications.
In this Table 1. The comparison of related studies conducted for the prediction of High-Risk Pregnancies only compared the related studies not the proposed method.
The paper presents a comparison of supervised machine learning methods, not clear, it is an exercise of data application with basic python algorithms.
The results of tables 3 to 8 are not clear. For example the table 8. Multi-Layer Perceptron Hyperparameter Tuning Results has a score of 98.18 but in table 9 it has a value of 91.86% for accuracy. What does score mean?
The assembled model of Self-Training model could be the contribution of the work, however its implementation is not clearly explained.
In my opinion the paper should be reconsidered after major revision (rewrite the implementation, results and conclusions)
Author Response
Point 1: The title of the paper does not correspond to the content of the paper. The work use only supervised machine learning algorithms.
Response 1: We used a Semi-Supervised Approach which requires the help of Supervised Machine Learning Algorithms. Although we still used Supervised ML Algorithms, there is still a need to augment it with specifically when data is scarce or weak. Updated the introduction to include why we used a Semi-Supervised Approach.
Point 2: The paper make use of data collected from local Health Centers and predict the risk of pregnancy complications.
Response 2: Yes, this is correct.
Point 3: In this Table 1. The comparison of related studies conducted for the prediction of High-Risk Pregnancies only compared the related studies not the proposed method.
Response 3: Updated the introduction to include the Machine Learning methodologies of Related Studies. Added a summary at the end of Related Studies.
Point 4: The paper presents a comparison of supervised machine learning methods, not clear, it is an exercise of data application with basic python algorithms.
Response 4: The paper compares supervised methods and the best one will be used as base classifier for the semi-supervised Self-Training model. Updated the Introduction and methodology to make it clear.
Point 5: The results of tables 3 to 8 are not clear. For example the table 8. Multi-Layer Perceptron Hyperparameter Tuning Results has a score of 98.18 but in table 9 it has a value of 91.86% for accuracy. What does score mean?
Response 5: Updated the Results section to include the source of data for each testing done. For clarification, Hyperparameter Tuning used Training Data, Comparative Analysis used Testing Data, and Self-Training model used Synthetic Data with 30% unlabeled data. These have been updated and explained in the methodology and results sections respectively.
Point 6: The assembled model of Self-Training model could be the contribution of the work, however its implementation is not clearly explained.
Response 6: Updated the Manuscript as per recommendation.
Point 7: In my opinion the paper should be reconsidered after major revision (rewrite the implementation, results and conclusions)
Response 7: Updated the Manuscript as per recommendation including adding a Discussion Section.

Reviewer 2 Report
The work proposed is timely, highly relevant, and of great interest in the research community. The paper has a good structure. I have the following comments, for authors to take note and further improve the quality of the manuscript.
1. At the end of Section 2, it would be good if all the findings of the literature review can be summarized, to highlight the research gap, before presenting the novelty/significance of the proposed work (line 47-49 needs to be rewritten/rephrased to emphasize on the significance of the work)
2. The references cited in Section 2 are from the years 2016 – 2020. Kindly include works from the recent years (2021 – 2022) to highlight the latest development of similar works.
3. Table 4 needs to be placed on a single page for better readability.
4. More discussion should be included to compare the results obtained with those reported in the literature. Is the best model found in your work consistent with the results reported from references 5-8? Justify and elaborate.
5. The 17 references cited in the work include 13 online sources. 3 conference proceedings, and only 1 journal article. The authors should cite more journal articles/publications.
Author Response
Point 1: At the end of Section 2, it would be good if all the findings of the literature review can be summarized, to highlight the research gap, before presenting the novelty/significance of the proposed work (line 47-49 needs to be rewritten/rephrased to emphasize on the significance of the work)
Response 1: Updated the Related Studies and added a summary at the end. Also, lines 47-49 was updated.
Point 2: The references cited in Section 2 are from the years 2016 – 2020. Kindly include works from the recent years (2021 – 2022) to highlight the latest development of similar works.
Response 2: Updated the Manuscript and added more 2022 references.
Point 3: Table 4 needs to be placed on a single page for better readability.
Response 3: Updated the Manuscript as per recommendation.
Point 4: More discussion should be included to compare the results obtained with those reported in the literature. Is the best model found in your work consistent with the results reported from references 5-8? Justify and elaborate.
Response 4: Updated the conclusion.
Point 5: The 17 references cited in the work include 13 online sources. 3 conference proceedings, and only 1 journal article. The authors should cite more journal articles/publications.
Response 5: Updated the Manuscript and added 9 more journal references and more 1 conference proceedings.

Round 2
Reviewer 1 Report
Comments were properly addressed.